# Machine Learning Approach to Predict Physical Properties of Polypropylene Composites: Application of MLR, DNN, and Random Forest to Industrial Data

**DOI:** 10.3390/polym14173500

**Published:** 2022-08-26

**Authors:** Chonghyo Joo, Hyundo Park, Hyukwon Kwon, Jongkoo Lim, Eunchul Shin, Hyungtae Cho, Junghwan Kim

**Affiliations:** 1Green Materials and Processes R&D Group, Korea Institute of Industrial Technology, 55, Jongga-ro, Jung-gu, Ulsan 44413, Korea; 2Department of Chemical and Biomolecular Engineering, Yonsei University, 50, Yonsei-ro, Seodaemun-gu, Seoul 03722, Korea; 3Research & Development Center, GS Caltex Corporation, 359, Expo-ro, Yuseon-gu, Daejeon 34122, Korea

**Keywords:** polypropylene composite, data categorization, machine learning, multiple linear regression, deep neural network, random forest

## Abstract

Manufacturing polypropylene (PP) composites to meet customers’ needs is difficult, time-consuming, and costly, owing to the ever-increasing diversity and complexity of the corresponding specifications and the trial-and-error method currently used to satisfy the required physical properties. To address this issue, we developed three models for predicting the physical properties of PP composites using three machine learning (ML) methods: multiple linear regression (MLR), deep neural network (DNN), and random forest (RF). Further, the industrial data of 811 recipes were acquired to verify the developed models. Data categorization was performed to account for the differences between data and the fact that different recipes require different materials. The three models were then deployed to predict the flexural strength (FS), melting index (MI), and tensile strength (TS) of the PP composites in nine case studies. The predictive performance results differed according to the physical properties of the composites. The FS and MI prediction models with MLR exhibited the highest R^2^ values of 0.9291 and 0.9406. The TS model with DNN exhibited the highest R^2^ value of 0.9587. The proposed models and study findings are useful for predicting the physical properties of PP composites for recipes and the development of new recipes with specific physical properties.

## 1. Introduction

Composites are mixtures of two or more materials and generally consist of a matrix that serves as a bonding material and a filler that serves as reinforcement. Polymerized composites may be used directly, and their physical properties can be obtained by implementing fillers or additives [1]. In particular, polypropylene (PP) composites are produced by polymerizing PP with other materials. PP is used in research on various composites, owing to its excellent moldability and mechanical properties.

PP composites are produced using the following steps. First, PP is mixed with stabilizers, fillers, and other additives. Second, the mixture is extruded and then packaged. In the first step, their physical properties vary depending on the type and quantity of materials [2,3]. Dikobe and Luyt investigated the morphology, as well as thermal and mechanical properties of blend composites. They found that wood powder has a higher affinity for ethylene vinyl acetate than PP. Ismail and Suryadiansyah found that PP with recycled rubber blends has high tensile strength and Young’s modulus, whereas PP with natural rubber blends has high elongation and stabilization torque. PP composites can be used in various fields depending on their physical properties, and the development of flexible recipes for their production is crucial. Hence, to quickly produce PP composites to meet demand, most chemical companies preemptively research new recipes. The development of recipes for manufacturing PP composites with new physical properties involves experimentation or research using empirical data. Recipe development is performed by adjusting the type and quantity of materials and testing the produced samples, which is a time-consuming and costly process. In particular, the melting index (MI) of PP composites needs to be measured because it is associated with the grade of the product; however, the analysis requires 2–4 h, and the cost is high [4].

The development time and cost can be reduced when the recipes of already-produced composites are applied to machine learning (ML). Jiang et al. predicted the MI of PP by applying variables such as the mix ratio and polymerization temperature, which represent the actual process conditions, to a relevant vector machine [5]. In like manner, predicting the physical properties using a prediction model can save time and costs because new PP composites can be produced quickly through a testing process only by omitting complex experimentation and research processes. The implementation of ML is reasonable because it is difficult for conventional theoretical models to identify the correlations between materials. 

However, because ML-based models are typically trained via random sampling rather than on every combination of materials, overfitting may occur when the model is applied to predict the properties of PP composites with their materials. Moreover, the optimal ML algorithms for the models depend on the correlations between the materials of the PP composites and their properties. Therefore, when an ML-based model for predicting the properties of PP composites with their materials is developed, numerous combinations of materials have to be considered to overcome the overfitting problem, and the optimal ML algorithm has to be selected for high performance of the model. 

To solve these problems, we developed and compared ML-based models for the flexible prediction of the physical properties of PP composites. In addition, we applied categorization as data preprocessing to consider the numerous combinations of materials. The physical properties used in this study are flexural strength (FS), MI, and tensile strength (TS), which are the primary physical properties to evaluate the specifications of PP composites, and the algorithms are multiple linear regression (MLR), deep neural network (DNN), and random forest (RF).

The novel and major contributions of this study are as follows:This study proposed and compared prediction models by training recipe-based data from a real PP composites plant.Categorization is applied as data preprocessing to overcome the overfitting issue.This is the first study to propose a suitable model according to physical properties.

Figure 1 shows a flowchart for the development of a new PP composite recipe. The demand for the various types of PP composites can be quickly satisfied by replacing experimentation and research with the developed models. To evaluate the model, the root mean squared error (RMSE) and R^2^ values of the actual and predicted data were compared. Then, for each physical property, the prediction model with the highest accuracy—the smallest RMSE and largest R^2^—was selected.

The remainder of this paper is organized as follows. Section 2 discusses the related work. Section 3 describes the three different ML models and presents the corresponding mathematical background. Section 4 details the development of the proposed models, including data preprocessing. Section 5 discusses the application and performance evaluation of the three models. Finally, Section 6 summarizes the important results and provides directions for future work.

## 2. Related Work

Unlike single linear regression analysis, MLR can be used when two or more variables influence the results. It is often deployed in data analysis, owing to its fast calculation speed and high accuracy. Liu et al. estimated the mercury content in the leaves of phragmites australis using MLR [6], and Ali et al. researched real-time wave height prediction [7]. Chen et al. proposed the MLR-based property prediction model using polymer data [8]. 

A DNN is a nonlinear regression method and is an advanced artificial neural network (ANN) algorithm. In an ANN, which was inspired by the neural networks in the brain, the nodes forming a network through the bonding of synapses solve problems through learning. Because an ANN can produce data for a specific situation using learned data, it has been applied to effectively solve many scientific and engineering problems that require experiments or measurements [9]. Its effects have been verified through several studies. Belalia Douma et al. predicted the properties of self-compacting concrete using an ANN [10]. They successfully developed an ANN-based prediction model to predict the slump flow, L-box ratio, V-funnel time, and the compressive strength of self-compacting concrete. Kuhe et al. applied ANNs to predict global solar radiation [11]. They used feed-forward backpropagation neural networks, radial basis function networks, and generalized regression neural networks as an ensemble solution. Ultimately, they successfully predicted solar radiation using their model and achieved an R2 of 0.998. Yılmaz et al. proposed the Pi-Sigma artificial networks for effective performance [12]. Tran et al. used ANNs to predict a variety of polymer properties [13]. In another study, de Sousa et al. verified the accuracy of ANNs in genome selection [14]. 

Although the ANN is appropriate for simple prediction because of the small number of hidden layers it possesses, its prediction ability is low in complex systems. Thus, a DNN with an increased number of hidden layers was developed. The DNN exhibits higher accuracy for complex prediction models than ANN, which uses one hidden layer [15]. Research on the DNN is still actively underway with respect to the prediction models using AI. Singh et al. developed a prediction model for road accidents using a DNN [16]. They used the official records and a dataset of 2680 accidents for data-driven modeling. They compared their DNN-based model to gene expression programming and random effect negative binomial models and showed that the DNN-based model achieved the highest performance. Lim et al. used a DNN to find the optimal blending ratio point of waste seashells [17]. They found the best ratio of waste seashells for SOx capture using a DNN-based surrogate model within a reasonable time. Qiao et al. proposed a DNN with a strengthening response sparsity for application to high-dimensional data [18]. They tried the strengthening response sparsity approach for deep learning to improve the sparse learning abilities and time complexity. They tested their model on the Fashion-MINIST dataset and successfully improved the DNN-based prediction model.

Another popular ML technique is ensemble learning, which uses several decision trees (DTs) for better prediction accuracy. For classification and regression analysis, the RF ensemble learning method is often used. Existing DT models can obtain the result closest to the goal in decision-making analysis, but they suffer from large variations in their accuracy. To overcome this disadvantage, the RF algorithm—using multiple DTs—was developed, which improved the generalization performance. Consequently, RF is used in various prediction models. For example, Franco et al. used RF to validate their data [19], and Shen et al. found that RF outperformed other commonly used classifiers, such as the support vector machine and DT [20].

A few other ML techniques, such as linear ridge regression (LRR), deep tensor neural network (DTNN), grammar variational autoencoder (GrammarVAE), and kernel ridge regression (KRR), can also be used for data-driven modeling [21,22]. However, in this study, MLR, DNN, and RF are used for the data-driven modeling of PP composites. 

## 3. Materials and Methods

To develop a prediction model for the physical properties of PP composites, this study used data composed of 811 recipes. The MLR, DNN, and RF algorithms were deployed for modeling three physical properties: the FS, MI, and TS, which are tested by the American Society for Testing and Materials (ASTM) D 790, 1238, and 638, respectively. Each algorithm is a representative algorithm for linear regression, nonlinear regression, and ensemble, and each has different characteristics. Thus, the case studies were performed for 9 cases in total, and the most appropriate prediction model for each physical property was determined.

### 3.1. Dataset

In total, 811 recipes were divided by the materials used and the mix ratios. A total of 90 types of materials were used to produce the PP composites: 41 types of PPs, 18 types of fillers, 22 types of rubbers, and 9 types of other additives. The recipes were obtained from GS Caltex Corporation, and because some of them are real PP products, specific materials were replaced with alphanumeric characters due to confidentiality issues. We have published some of the encoded data as Appendix A. The raw data are listed in Table 1 and illustrated in Figure 2, where P, F, R, and OTH denote PP, filler, rubber, and others, respectively. The accompanying numbers are given to distinguish the different materials. Numbers 0 to 100 correspond to the wt% values of the material content included in the recipes of the PP composites. Figure 2 shows that there are some 0 wt% in the material data and blanks in the property data. Table 1 shows the multiplicity of materials that can be used for composites. This means that the dataset is incomplete for data-driven modeling.

FS is the maximum force that can bend a material without permanently distorting or damaging it. In other words, it is the maximum stress acting on an external surface that is subject to tensile stress at the moment of fracture. As a representative mechanical property of PP composites [23,24], the measure of FS, σF, is calculated using Equation (1), where L, b, and d denote the length, width, and height of the object, respectively; and *F* denotes the vertical force acting on the object. Figure 3 illustrates the measurement of the FS.
(1)$$\sigma F=\frac{3FL}{2bd^{2}}$$

The melting properties of polymers are important because they are directly related to the processability of products [25,26]. The MI is a representative parameter of the melting property. It indicates the weight of the resin flowing through a capillary for 10 min at a constant load and temperature. The measurement of the MI is illustrated in Figure 4. The extrusion process is executed by the piston of the material in the melted state under a high temperature. The die, test sample, and weight of the resin flowing for 10 min correspond to the capillary, molten resin, and measured the MI, respectively. The factors that have the largest effect on the MI are the molecular weight and its distribution. Because these two values are difficult to measure, the mean molecular weight of the polymer is sometimes estimated based on the MI.

TS refers to the maximum stress when the material is fractured by a tensile load. TS serves as a test for measuring various characteristics of a material under tension and is the most general item among the mechanical property tests of plastics. It can be mathematically calculated using Equation (2), where *A* and *F* denote the cross-sectional area of the specimen and maximum force applied to the specimen, respectively. Figure 5 shows the strain–stress curve, where *σ*_s_ corresponds to TS. As the strain, *ε*, of the material under a force increases, the stress, *σ*, inside the material increases and then decreases, resulting in a fracture of the material:(2)$$\sigma_{S}=\frac{\mathrm{Ultimate}\;\mathrm{Force}}{\mathrm{Original}\;\mathrm{cross}-\mathrm{ectional}\;\mathrm{area}}=\frac{F}{A}$$

### 3.2. Multiple Linear Regression

MLR comprises multiple independent variables. Hence, it is different from the single linear regression analysis, which only describes the relationship between one independent variable *x* and one dependent variable *y*. The basic mathematical expressions of single and multiple linear regressions are shown in Equations (3) and (4), respectively, where m and e denote the regression coefficient and error term, respectively.
*y* = *m*_0_ + *mx* + *e*(3)
*y*_i_ = *m*_0_ + *m*_1_*x*_i1_ + *m*_2_*x*_i2_ + … + *m*_k_*x*_ik_ + *e*_i_, *i* = 1, 2,…, *n*(4)

To use MLR, one or more independent values must be inputted into Equation (4). This requires the number of variables, *n*, used in the prediction to be equal to or greater than the number of independent variables, k.

### 3.3. Deep Neural Network

Unlike MLR, the DNN is a nonlinear regression. The DNN comprises hidden layers that analyze the relationship between the input and output values based on mathematical and statistical methods. Unlike ANN, which has one hidden layer, the DNN has multiple hidden layers (three or more) between the input and output layers [27]. The DNN complex prediction model yields a high accuracy when handling data that are not appropriate for linear analysis, and many researchers have reported excellent predictive power and successful application cases [28,29,30,31].

The DNN deployed in this study uses three hidden layers, as shown in Figure 6. When an input value enters the hidden layer, the weight, w, and bias, b, of each input are determined in the synapse between the input and hidden layers. While the determined values pass through the hidden layer, the output is obtained through an activation function that converts the sum of input signals into output signals. Because the DNN in this study consists of three hidden layers, this process is repeated three times before the y value is called. A detailed mechanism is illustrated in Figure 7 [32]. If there are several inputs and nodes in the input and hidden layers, respectively, the mechanism would be more complicated, as shown in Figure 7. By forming several paths from the input layer to the output layer, the DNN can express the relation between input and output variables more precisely.

There are many types of activation functions in a DNN, including the sigmoid and tanh. This study uses the rectified linear unit (ReLU) function. The ReLU function outputs the input if the input exceeds zero or outputs zero if the input is equal to or less than zero, as deduced by Equation (5) and Figure 8 [33]. Thus, a low computing cost, simple implementation, and fast computing speed are achieved [34].
ReLU(x) = max(0,x)(5)

### 3.4. Random Forest

The RF is an ensemble learning method that implements multiple DTs and optimally classifies data using the training data. Although the DT is effective for data classification, its performance varies significantly, and it contains different errors, mainly because the generated DT varies depending on the given training data. An RF algorithm was developed to overcome this drawback.

Figure 9 shows an RF algorithm that is made of multiple DTs, where each DT has slightly different characteristics owing to randomness. These characteristics improve the generalization performance of the algorithm. Meanwhile, when the data are trained to develop a prediction model using the RF algorithm, the stability, or accuracy of the model may be reduced owing to the trees. Hence, bagging is mainly used to sample the training data [35]. Bagging is an abbreviation for bootstrap aggregating, and it refers to the process of generating a dataset of the same size as the existing dataset by allowing duplicates in the training data. It improves the performance of RF by reducing variance while maintaining the bias of the trees [36]. If bagging is not used, underfitting or overfitting may be seen, owing to the high bias or high variance, respectively. This could further affect the performance of the prediction model.

## 4. Machine Learning Model Development

### 4.1. Overview of Model Development

An overview of the model development process is shown as a flowchart in Figure 10. First, the blanks were removed from the 811 PP composite recipes, and the data were divided into FS, MI, and TS data. Then, after categorizing the classified data, data split was performed at a ratio of approximately 7:1:2 for the training, validation, and testing datasets. The training and validation data were standardized by the Z-score, and the corresponding processes of the three prediction models were performed. Subsequently, the most suitable prediction model for each physical property was selected by comparing the results obtained using the testing data with those of the prediction model and calculating the RMSE and R^2^.

### 4.2. Data Categorization

The general ML technique randomly samples data and classifies them into training, validation, and testing datasets. However, underfitting, which refers to a state of not approaching the decision boundary because the number of data is too small or the training is not performed properly, occurred when the prediction model was created by random sampling because the data used in this study consisted of 90 types of composite materials, whereas the total number of recipes was 811. This issue must be resolved during the development of a prediction model because it lowers the achieved accuracy. Furthermore, although 90 types of materials are considered in this study, no composites comprise all 90 types simultaneously. Thus, it is difficult to identify the correlations between the materials and physical properties. Consequently, the composites were classified by the presence or absence of materials before creating the prediction model; this process was defined as ‘categorization’. When the data are classified in advance through categorization, underfitting can be avoided by training all types of data, and the correlations between the materials and physical properties can be identified through the analysis of the physical properties by type. The categorization process is illustrated in Figure 11 [37]; the same annotations as those in Figure 2 are used. The details of categorization are shown in Appendix A.

A code we made, comprising T and F, was assigned to every recipe, depending on the presence or absence of materials. The recipes with identical codes were classified as the same type, and a total of 496 types were identified from 811 PP composite recipes. The amount of data and the number of types for each physical property, excluding the blank data, are shown in Table 2. The numbers of recipes and recipe types of FS data, excluding the blank data, were 803 and 494, respectively. The amount of data related to FS was the largest among the three properties. Next, the TS data, excluding the blank data, consisted of 801 recipes and 493 recipes. The MI data, excluding the blank data, had the smallest numbers, with 480 recipes and 339 recipes.

### 4.3. Preprocessing

After categorization, the data were classified into training, validation, and testing datasets. To improve the accuracy of the prediction model, the training dataset must include all types of data. Thus, as shown in Table 3, certain rules were applied to prevent underfitting and overfitting, and the classified data were randomly selected and divided based on a 7:1:2 ratio, approximately, for the training, validation, and testing datasets. When the categorized data were analyzed, one type included 1 to 12 recipes. For example, for a 12-recipe type, 9 recipes were classified into the training set, 1 recipe into the validation set, and 2 recipes into the testing set. Table 4 shows the exact ratios of the data classified by applying the abovementioned rules. 

To reduce the influence of the outliers in the preprocessing process, a Z-score standardization method was used. Because the data of the PP composites include many outliers, Z-score standardization was applied to reduce the impact of the outliers. The Z-score standardization is expressed in Equation (6):(6)$$Z=\frac{\left(x-\bar{x}\right)}{\sigma }$$
where Z denotes the standardized score, x denotes the original score, $\bar{x}$ denotes the sample mean, and  σ denotes the standard deviation of the samples. If the absolute value of Z exceeds 3, it is considered an outlier. 

The standardization of the TS data is shown in Figure 12 as an example. The numbers on the x-axis refer to the recipe numbers introduced for distinguishing the PP composites. Figure 12a shows the TS data distribution chart before standardization. Figure 12b,c present the TS data distribution chart immediately after standardization for different y-axis scales. It can be seen that, after Z-score standardization, outliers gather near Z = 0, thus diminishing the differences with the mean TS. Subsequently, the performance of the prediction model is improved by reducing the numerical influence of outliers. Because the outlier data in this study represent the values of actual PP composites, they were considered meaningful data. Therefore, only the standardization was performed without removing outliers. After the model learned the standardized data, the results were restored to the original scale and evaluated accordingly.

### 4.4. Modeling 

The input variables that were used to develop the model for predicting the physical properties were set as the ratio (wt%) of the 41 types of PP (P001–P041), 18 types of filler (F001–F018), 22 types of rubber (R001–R022), and 9 types of other additives (OTH1–OTH9) that constituted the composites. The output variables to be predicted were set as the physical properties (FS, MI, and TS). Because every algorithm is used to predict the properties, the models are regressors, which predict the continuous values. Table 5 shows a list of hyperparameters that need to be set. All the hyperparameters are tuned heuristically using the validation dataset in this study. After training the model with different hyperparameters, the performances were compared using the validation dataset. Then, the hyperparameters which showed the highest R^2^ were selected as the final hyperparameters in this study. The model performances on the training and validation datasets are shown in Table 6.

In MLR, the intercept fitting was set to “True”, and in the DNN, three hidden layers were used, which comprised 45 nodes in the first layer and 10 in the second and last layers. In addition, Adam was used as the optimizer, and the batch size was set to 3. For the epochs, the earlystopping function, which stops the training process in a specific condition, was used. For the RF, 100 estimators were used, and the maximum depth and minimum samples leaf were set to 10 and 3, respectively. We developed our models in Python 3.7.7 using scikit-learn 1.0 and Tensorflow 1.15.

## 5. Results and Discussion

### 5.1. Validation of Categorization

To verify the validity of the categorization process, parity plots for the MLR-predicted values and actual values, before and after categorization, are shown in Figure 13. Before categorization, 80% of the data were used for the training data, and 20% for testing, through random sampling. In the left graphs of Figure 13, the predicted values that are significantly different from the actual values are marked by red squares. Large errors appear because the amount of data used was small, and the physical properties were predicted using the untrained data type. Owing to the large errors, the other predicted values were represented by 0 in the graph. To solve this problem, the physical properties were predicted again using MLR after data categorization; the results are shown in the right graphs of Figure 13. The solid red lines indicate a perfect agreement between the actual and predicted values. A small distance between the blue dots and the red line indicates a high-performing prediction. It was found that the differences between the actual and predicted values were small when the data, after categorization, were used. The outline change in the graph confirms that the errors, due to the insufficient number of training data and inaccurate training, were eliminated through categorization. Before categorization, the training dataset, and test dataset may have recipes including different materials, which signifies “input variable inconsistency.” After categorization, the training dataset and test dataset have recipes that include the same materials. Therefore, categorization reduces the error of variable inconsistency.

### 5.2. Comparison of Prediction Model Performance 

The results of the MLR, DNN, and RF prediction models for each physical property are illustrated in Figure 14. The solid red line signifies the perfect agreement between the actual and predicted values. A small distance between the blue dots and the red line indicates a high-performing prediction model. Examining the MLR and DNN prediction results of FS indicates that the distances between the red lines and blue dots are generally small. In contrast, the RF prediction result of FS shows that, regardless of the size of the FS values, the accuracy was lower than that of the other two models. Similar behavior was observed for the TS prediction results. Further, the prediction results of the MI show that the degree of aggregation of the blue dots near the red line was low in the RF model and high in the MLR and DNN models. Consequently, MLR achieved excellent accuracy in the FS and MI modeling, whereas the DNN was the most effective for TS prediction.

To select a prediction model with excellent performance for each physical property, the RMSE and R^2^ between the actual and predicted values were calculated using Equations (7) and (8), respectively. The results are summarized in Table 7.
(7)$$\mathrm{RMSE}=\sqrt{\frac{\sum_{i=1}^{N}{\left(S_{\mathrm{im}}-S_{\mathrm{ip}}\right)}^{2}}{N}}$$
(8)$$R^{2}=1-\frac{\sum_{i=1}^{N}{\left(S_{\mathrm{im}}-S_{\mathrm{ip}}\right)}^{2}}{\sum_{i=1}^{N}{\left(S_{\mathrm{im}}-{\bar{S}}_{\mathrm{ip}}\right)}^{2}}$$

In the above equations, *N* denotes the number of data units, *S*_im_ denotes the actual values, and *S*_ip_ denotes the predicted values of the ML model. The RMSE value, calculated using Equation (7), represents the degree of error of the prediction model. Thus, the RMSE values closer to zero reflect a high prediction performance. The R^2^ value, calculated using Equation (8), is an indicator that quantifies the relative performance of the prediction model by comparing the predicted value to the actual value. It is difficult to directly determine the performance solely based on the RMSE because the value differs from the data scale. Contrarily, R^2^ allows intuitive interpretation because it represents the relative performance. In the prediction performance evaluation, an R^2^ value exceeding 0.7 is generally considered a good performance. Based on Table 7, for the FS modeling, MLR showed the best prediction performance with an RMSE of 8.3122 and R^2^ of 0.9291, followed by the DNN and RF. MLR and the DNN exhibited similar prediction performances, whereas the RF model achieved a lower prediction performance than the other two models, with an RMSE of 9.9609 and R^2^ of 0.8981. For the MI modeling, MLR showed the best prediction performance, with an RMSE of 2.4072 and R^2^ of 0.9406, followed by the DNN and RF. Although the trends between the FS and MI modeling were similar, the prediction performance for the MI was higher than that for the FS. In contrast, for the TS modeling, the DNN showed the best prediction performance, with an RMSE of 4.9358 and R^2^ of 0.9587, followed by RF and MLR. Unlike the prediction performances for the FS and MI, MLR achieved the lowest R^2^ for the TS. This suggests that MLR is more appropriate for predicting the FS and MI of PP composites, whereas the nonlinear regression models (DNN and RF) are more suitable for predicting the TS. Therefore, different prediction performances can be obtained depending on the physical property, even if the same prediction model is used and vice versa. Hence, to accurately predict physical properties, a prediction model that is most appropriate for each physical property must be developed.

Additionally, the relative variable importance (RVI) for each property is calculated to analyze the sensitivity. The RVI can be calculated using the coefficients of MLR, as shown in Equation (9). The coefficient quantifies the impact of the material on the properties in MLR, and the quantified values indicate the sensitivity of each property.
(9)$$RVI_{i}=\frac{Coef_{i}}{\max\left|Coef\right|-\min\left|Coef\right|}$$
where $RVI_{i}$ is the relative variable importance of the i^th^ material in the recipe and $Coef_{i}$ is the coefficient of the i^th^ material in the MLR equation. Figure 15 shows five materials with high RVIs for the MI, FS, and TS. Figure 15a shows four types of rubber that possess a high RVI for the MI. Therefore, these four types of rubber can be used to control the MI of the PP composites. In Figure 15b,c, it can be seen that filler and other additives contribute more to the FS and TS than PP and rubbers. Therefore, it is considered that filler and other additives can control the FS and TS of PP composite.

## 6. Conclusions

In this study, models were developed for the prediction of the physical properties of PP composites using the MLR, DNN, and RF algorithms. Typically, prediction models are trained via random data sampling, which results in poor performance. To address this issue, we categorized the data during preprocessing for the first time. The prediction model using the categorized data showed a higher performance than the prediction model using the uncategorized data. Furthermore, it was verified through recipes that categorization is essential for developing a prediction model for the physical properties of PP composites. The performance of the prediction model was further enhanced by the Z-score standardization to reduce the influence of outliers during model training. When the performances of the prediction models for the three physical properties of the FS, MI, and TS were compared, the MLR, DNN, and RF showed R^2^ values of 0.8 or higher, thus satisfying the standard value of 0.7 for using the prediction model. Among them, MLR showed the highest performance in the FS and MI prediction, with RMSE and R^2^ values of 8.3122 and 0.9291 and 2.4072 and 0.9406, respectively. The DNN achieved the highest performance in the TS prediction, with RMSE and R^2^ values of 4.9358 and 0.9587, respectively. Based on the results, we verified that even when the same prediction model is used, its performance varies depending on the physical property to be predicted. Therefore, the appropriate prediction model can predict each physical property of the PP composites with high accuracy. Thus, new recipes for PP composites with the desired physical properties can be developed by the results obtained in this study.

## Figures and Tables

**Figure 1 polymers-14-03500-f001:**
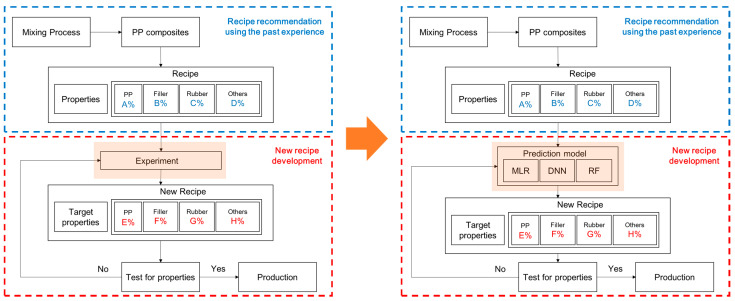
Flowcharts for developing recipes for polypropylene (PP) composites.

**Figure 2 polymers-14-03500-f002:**
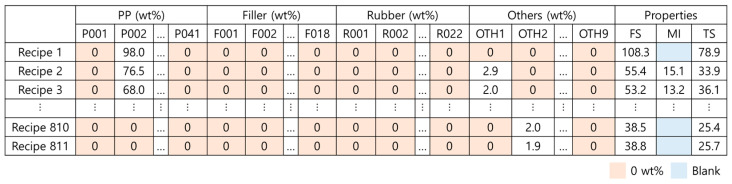
Raw data of PP composite recipes, including 90 types of material and FS, MI, and TS.

**Figure 3 polymers-14-03500-f003:**
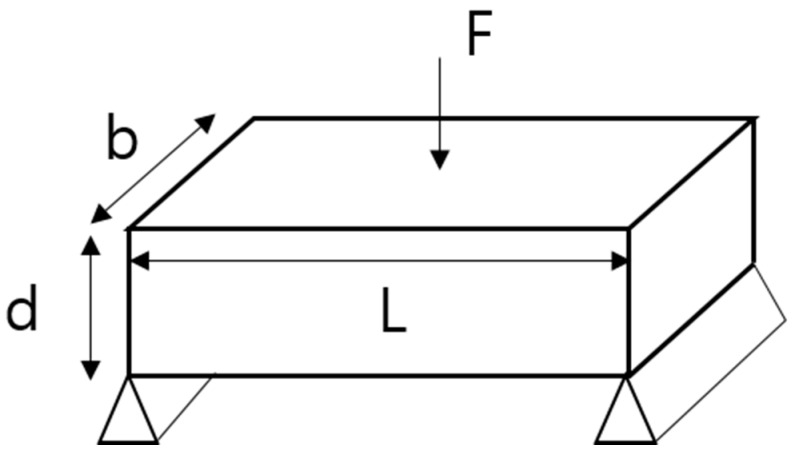
Schematic representation of FS acting on an object.

**Figure 4 polymers-14-03500-f004:**
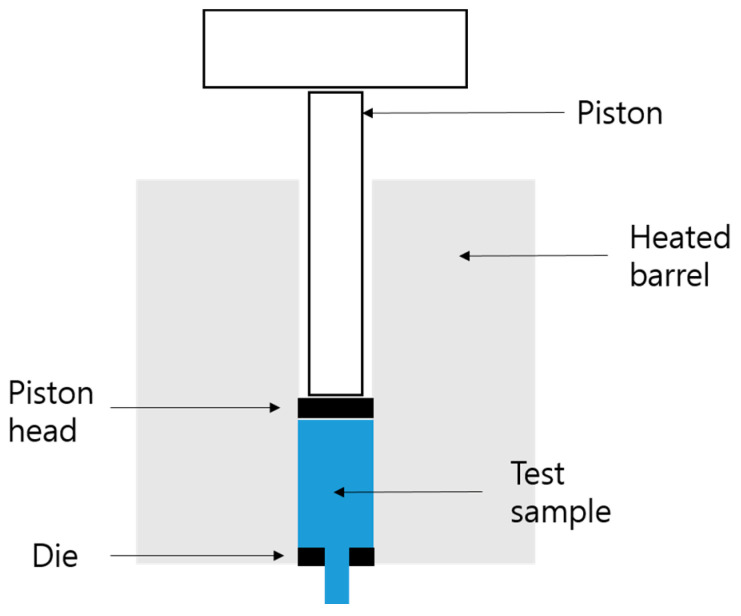
Experimental setup for measuring the MI.

**Figure 5 polymers-14-03500-f005:**
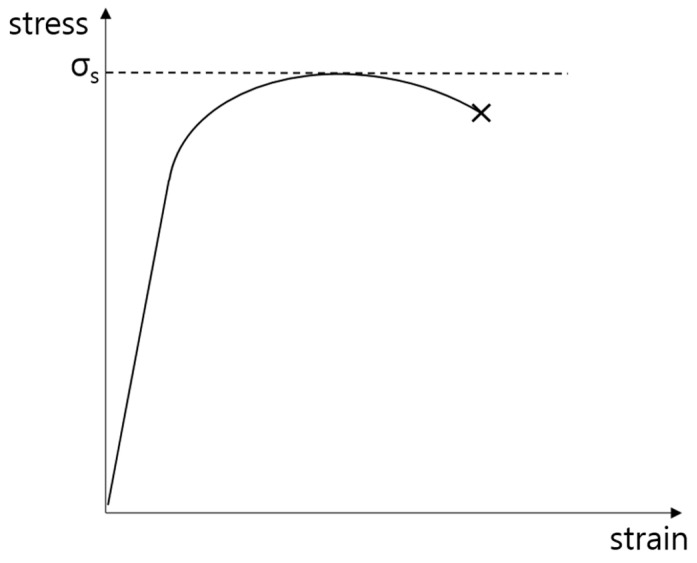
Strain–stress curve.

**Figure 6 polymers-14-03500-f006:**
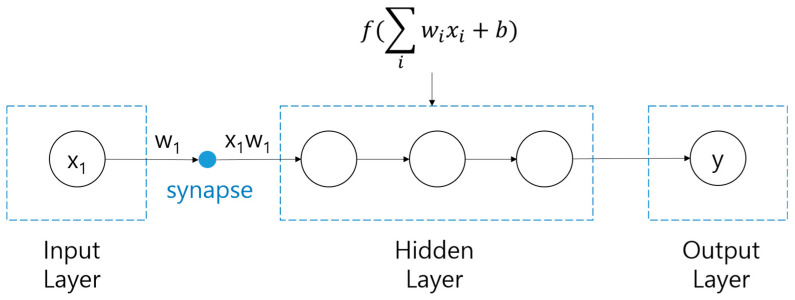
Schematic representation of DNN algorithm.

**Figure 7 polymers-14-03500-f007:**
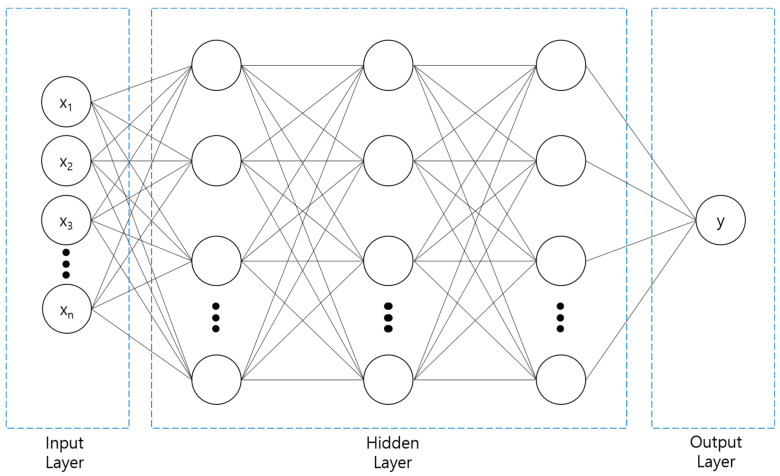
Illustration of a DNN with three hidden layers.

**Figure 8 polymers-14-03500-f008:**
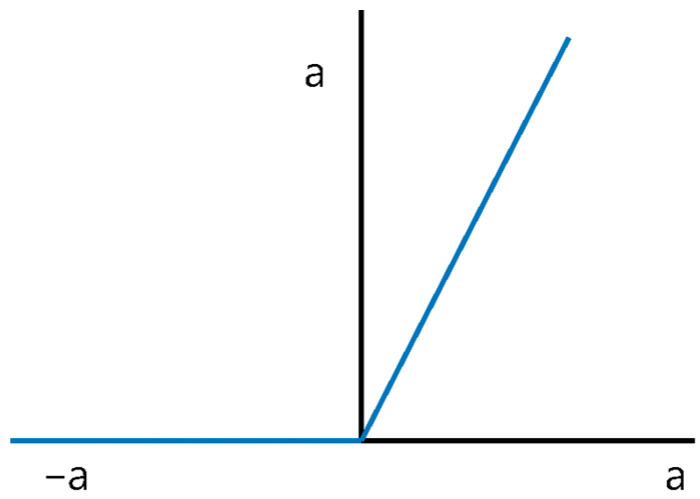
Graph of the ReLU function.

**Figure 9 polymers-14-03500-f009:**
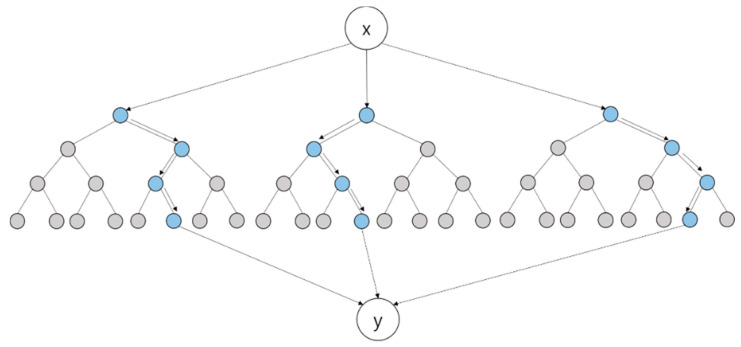
Schematic representation of random forest algorithm.

**Figure 10 polymers-14-03500-f010:**
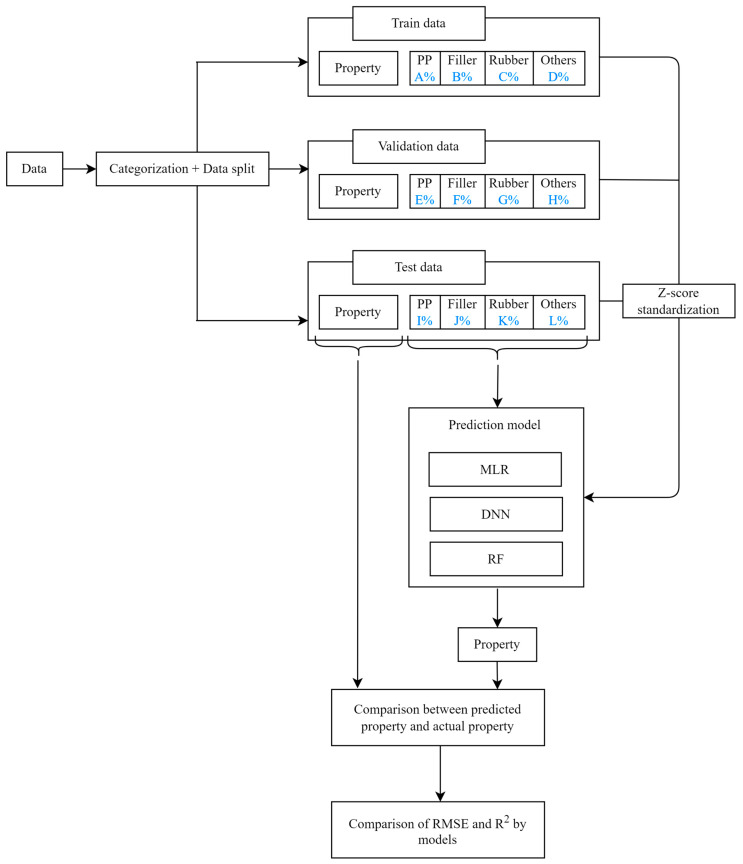
Flowchart of model development.

**Figure 11 polymers-14-03500-f011:**
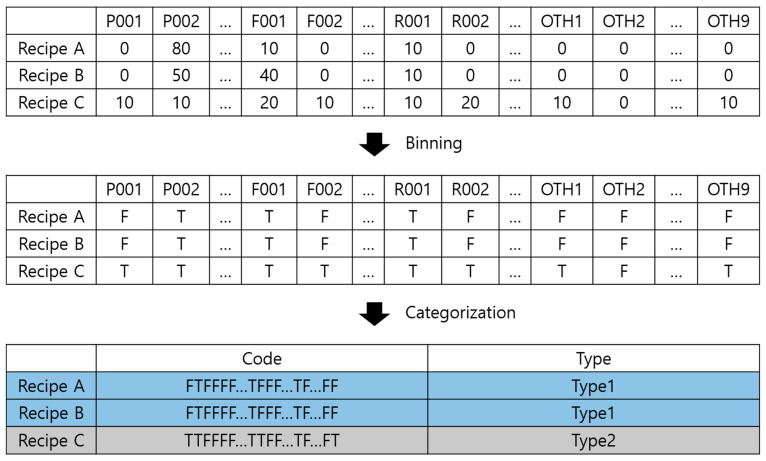
Process of composite categorization based on the presence or absence of materials.

**Figure 12 polymers-14-03500-f012:**
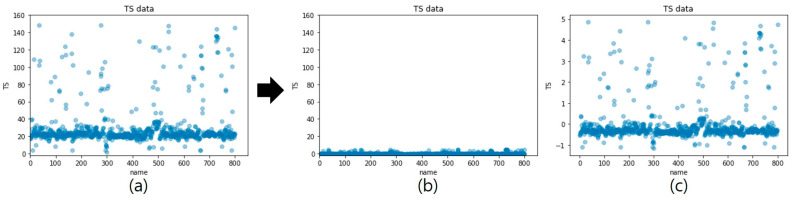
TS data distribution (**a**) before Z-score standardization, (**b**) after Z-score standardization, and (**c**) Z-score standardization for different y-axis scales.

**Figure 13 polymers-14-03500-f013:**
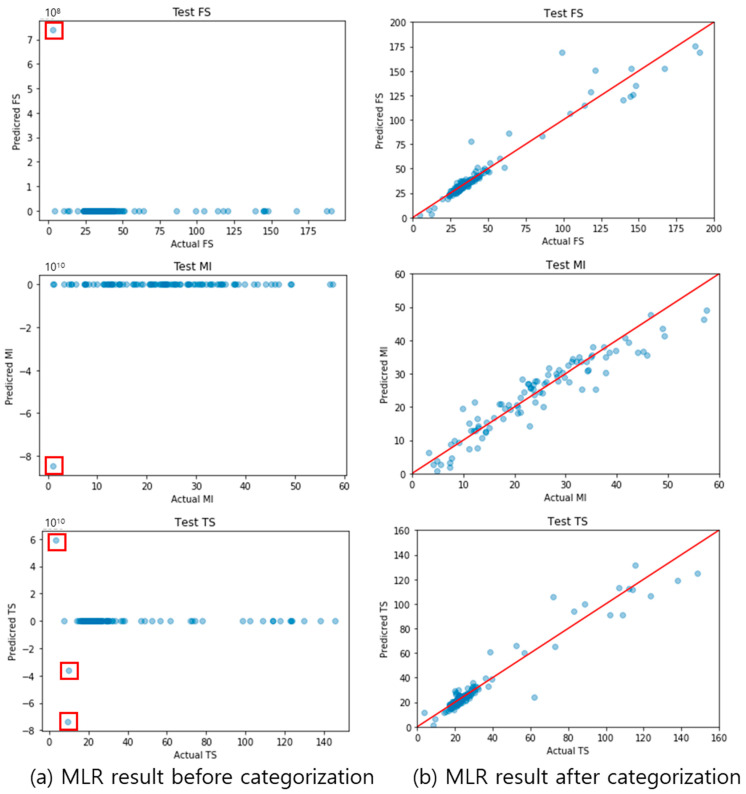
Parity plots of MLR results using pre-categorization data (**a**) and post-categorization data (**b**); the red square indicates a large error point between the actual and predicted value, and the red line is y = x (where x is the actual value and y is the predicted value).

**Figure 14 polymers-14-03500-f014:**
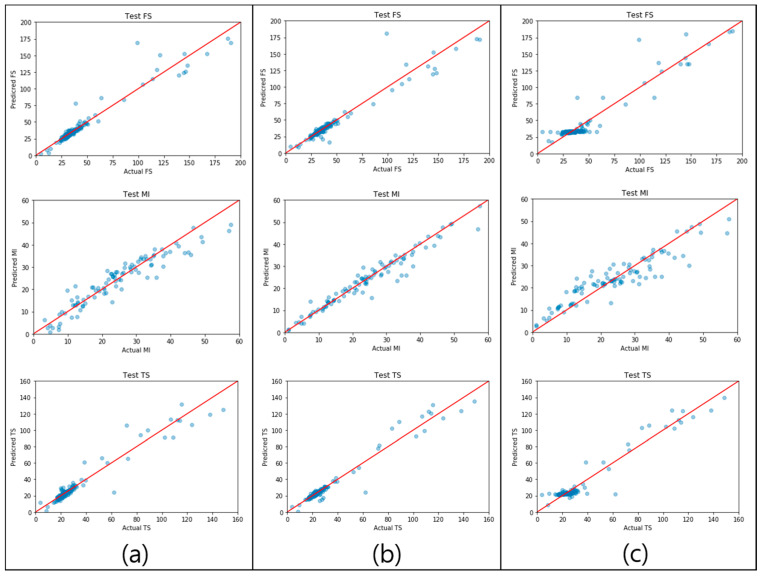
Parity plots of (**a**) MLR, (**b**) DNN, and (**c**) RF for the three physical properties; the red line is y = x (where x is the actual value and y is the predicted value).

**Figure 15 polymers-14-03500-f015:**
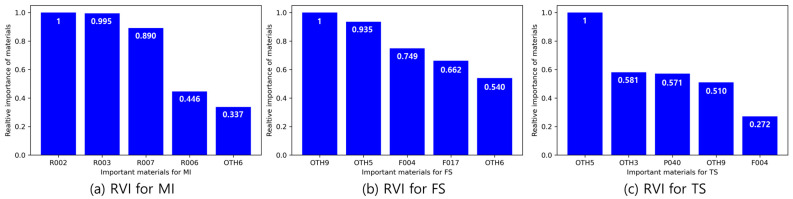
RVIs of (**a**) MI, (**b**) FS, and (**c**) TS for three physical properties.

**Table 1 polymers-14-03500-t001:** The number of different materials used per PP composite component.

Component	PP	Filler	Rubber	Others	Total
Number of materials	41	18	22	9	90

**Table 2 polymers-14-03500-t002:** Amount of data for the studied physical properties.

	Number of Recipes	Number of Composite Types
FS	811 → 803	496 → 494
MI	811 → 480	496 → 339
TS	811 → 801	496 → 493

**Table 3 polymers-14-03500-t003:** Number of datasets after data classification.

Number of Recipes in a Type	Number of Training Datasets	Number of Validation Datasets	Number of Testing Datasets
1	1	0	0
2	1	0	1
3	1	1	1
4	2	1	1
5	3	1	1
6	3	1	2
7	4	1	2
8	5	1	2
9	6	1	2
10	7	1	2
11	8	1	2
12	9	1	2

**Table 4 polymers-14-03500-t004:** Percentage of datasets per physical property.

Property	Training Data	Validation Data	Testing Data
FS	71.6%	8.3%	20.1%
MI	73.6%	6%	20.4%
TS	71.2%	8.4%	20.4%

**Table 5 polymers-14-03500-t005:** Values of the hyperparameters set for DNN and RF.

	Hyperparameter		Value
MLR	Intercept fitting		True
DNN	Type		Regressor
	Number of nodes in hidden layer1		45
	Number of nodes in hidden layer2		10
	Number of nodes in hidden layer3		10
	Optimizer		Adam
	Learning rate		0.001
	Batch size		3
	Loss		mean_squared _error
	Epochs		earlystopping
	earlystopping	monitor	val_loss
		patience	10
		verbose	1
RF	Type		Regressor
	Number of estimators		100
	Bootstrap		True
	Max depth		10
	Min samples leaf		3

**Table 6 polymers-14-03500-t006:** Values of the hyperparameters set for DNN and RF.

Algorithm	Property	Training Data	Validation Data
MLR	FS	0.9717	0.9793
	MI	0.9193	0.9426
	TS	0.9559	0.9445
DNN	FS	0.9796	0.9850
	MI	0.9854	0.9321
	TS	0.9801	0.9472
RF	FS	0.9852	0.9862
	MI	0.9607	0.8904
	TS	0.9837	0.9585

**Table 7 polymers-14-03500-t007:** RMSE and R^2^ by prediction model for the three physical properties.

	FS		MI		TS	
	RMSE	R^2^	RMSE	R^2^	RMSE	R^2^
MLR	8.3122	0.9291	2.4072	0.9406	6.2689	0.9334
DNN	8.5404	0.9254	3.3413	0.9297	4.9358	0.9587
RF	9.9609	0.8981	4.9732	0.8442	5.9648	0.9397

## Data Availability

Not applicable.

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
