# Peer review of "Machine Learning Approach to Predict Physical Properties of Polypropylene Composites: Application of MLR, DNN, and Random Forest to Industrial Data"

_polymers, 2022, doi:10.3390/polym14173500_

Round 1

Reviewer 1 Report

See attached PDF.

Reviewer 2 Report

In the current report, Joo and co-authors have used three machine methods (MLR, DNN, and Random Forest) to train, validate and predict the physical properties of polypropylene composites. The paper is expected to have broad interests to the researchers in the close field, however, the structure and the interpretation of the paper must be improved before I can accept it to be published on Polymers. My concerns are in the following:

Major points:

1.     As a simulation/modeling study, the simulation details, including software, packages (versions) and parameters should be provided for reproducing the work. However, in the current report, none of them (maybe except some of the parameters) were provided.

2.     I suggest that all parts before section 4.2 need to be simplified, the paper contains 18 pages (not including the reference), the core parts (results and discussions) only appear after 12 pages, I have to emphasis that it looks like I am reading a machine learning textbook with examples rather than a research paper.

3.     The authors claimed that they had developing new models for predicting physical properties of PP composites, however, I didn’t see any method development. Are you really developing new methods? How does the performance look like when comparing to existing methods? Where are the codes the authors have developed? For example: the paper published on [Polymers 2021, 13, 1898. https://doi.org/10.3390/ polym13111898] has conducted a machine learning method to predict the glass transition temperature of the polymer materials, the readers can easily download their source code and reproduce their work, this is important!

4.     Continue the issue with Question 3, the interpretations of the results are shallow, missing necessary comparisons with existing methods in the literature.

5.     All figure captions for the results and discussions must be improved, the reader should have the capability to understand the presentation of the figure with the figure cation and there is no need for them to find information in the main text. For example, the red square and the red line must be explicitly mentioned in the figure caption in Figure 15.

Minor points:

1.     Line 196: “where F and A denote the cross-sectional area of the specimen and maximum force applied to the specimen” should be “where F and A denote the maximum force applied to the specimen and the cross-sectional area of the specimen”.

2.     From line 364, the numbering of the tables is incorrect.

3.     Check the format and the presentation of Figure 14, are you taking snapshots rather than using output directly from Jupyter Lab/Notebook? The resolution of the figure is not good enough and there are strange lines/borders surrounding the figures.

4.     Should the Y-axis for Figure 14(c) be “Z”?

Reviewer 3 Report

This has some comments for the author, who may be unfamiliar with data processing or model application in general. But this is still a manuscript with publication potential.

1. Why use flexural strength (FS), MI, and tensile strength (TS) for physical properties?

2. Why machine learning methods use multiple linear regression (MLR), deep neural network (DNN), and random forest (RF)?

3. The raw data is incomplete, whether there has been data preprocessing, such as data cleaning. In addition, the author cuts the data set according to physical properties, does it make sense? It seems that the author does not want the data to be reduced after preprocessing, so it is predicted separately. However, this does not allow a comprehensive comparison to be performed.

4. "First, blanks were removed from the 811 PP composite recipes," how many blanks were removed? There are many places in the manuscript where numerical values ​​are mentioned, which the authors do not clearly state. Please improve together.

5. Use a lot of coding methods for classification, why not define i-th recipes directly. The author's description and definitions for the dataset section are very confusing and need to be revised or presented more clearly in a diagram or table.

6. "Table 3. Percentage of datasets per physical property" Usually random forests don't use a validation set? Does the author use it? The table can then subdivide the proportion of datasets by model type.

7. "Table 4. Values ​​of the hyperparameters set for DNN and RF. "Why is there no parameter design for MLR?

8. Is the author's proposed model a classifier or a regressor? (Explain in Table 4 and text)

9. Please clearly define the input and output parameters of the model.

10. The data set in Section 4 is poorly described, resulting in unclear meaning of the narrative in Section 5.1.

Round 2

Reviewer 1 Report

The authors have addressed all comments provided by me after the first review. The writing could be polished a little more w.r.t to sentence structure and grammar.

The only remaining issue is that authors are unable to provide details regarding the materials used in preparing composite recipes due to confidentiality required by industry collaborator. I feel such details are important to include in the manuscript for completeness and reproducibility. But I'll let the editor decide if these restrictions are acceptable to the journal.

Reviewer 2 Report

Based on the authors’ reply, I still have several concerns before the report can be accepted and published on Polymers.

1.     From the authors’ reply: “The algorithms we used to develop a property prediction model are commonly used algorithms. However, a new method for developing our model is the categorization, which is performed in data preprocessing.” It seems that the main contribution of the study is the development of the categorization, however, I cannot see related motivations or highlights from the Introduction. The authors should devote more effort on why categorization is important, and emphases how would categorization help to solve the over- or under-fitting problem.

2.     I am not convinced by the authors’ reply regarding the source code and details of the method implementation, as quoted below: “We understand that many papers have studied development-related code. However, at present, the code related to the model developed cannot be disclosed for corporate security. If this problem is resolved, we will open it.” The proper procedure for publishing the method is to submit your full implementation (as a computational study), the referee and the reader should be able to validate the implementation in the paper. If the authors think they have corporate concerns, they should publish the method later, or, as an alternative way, the authors could only submit sample code and related data (that’s what many computational study did). If the referee cannot even see and validate the implementation, there would be potential risks for the agreement of publishing the paper. Remember, the study is not about using widely used programs, which could be reproduced easily.

3.     The new Figure 15 still has the strange border problem.

4.     Figure cation of Figure 11 misses the full stop symbol.

Reviewer 3 Report

According to the author's previous point-to-point responses, the results are not satisfactory, with only a few reasonable corrections. Based on the revised manuscript and the authors' responses, more issues need to be addressed.

1. The authors make three contributions, the last of which they claim is the first study to come up with a model that fits....... I have seen similar research articles in different countries and journals. In contrast, the application of this manuscript is not novel.

2. According to the architecture diagram in Figure 1, I would like to ask the author at what time should the prediction task be executed? Before or after the mixing process? From Figure 1, it is not clear what the purpose of the model is.

3. What type of research is this manuscript? Is it about data science? Or about material development? The authors do not discuss either or both in depth. If this is data science research, the authors did not handle it well based on the previous review comments. If this is a material development study, the authors have not been seen discussing material formulations.

4. I feel unbelievable for the author's statement that MLR has no parameters to set.

5. The description of input parameters and output parameters in Section 4.4 does not match Figure 11.

6. In Table 5, what is the number of neurons in the input layer and output layer of DNN? What is the substantial difference between this design and the design of MLR? Why is the Min samples leaf in RF designed to 3?

7. If the author divides the data into training set, validation set and test set, please show the performance of the model in different data sets respectively.

Round 3

Reviewer 1 Report

 The only remaining issue is that authors are unable to provide details regarding the materials used in preparing composite recipes due to confidentiality required by industry collaborator. I feel such details are important to include in the manuscript for completeness and reproducibility. But I'll let the editor decide if these restrictions are acceptable to the journal. 

Reviewer 2 Report

The full stop symbol is missing at the end of the Figure 11's caption.

Reviewer 3 Report

Thank you to the author for patiently revising each review result, there are last important issues that need to be revised. With these issues resolved, the manuscript is almost ready for publication.

1. The authors use unpublished recipes data as the novelty of the study. But as far as research is concerned, studies that do not follow the authors' method to achieve the same results should not be reviewed and published. Such research work is unfair and unreliable. The authors emphasize that data preprocessing improves the performance of the model. But the preprocessing method is questionable. Therefore, the authors must be careful to deal with the negative effects of the recipes on the study. The authors must address this issue in Section 1.

2. The encoding in Table 11 depends on the presence or absence of the material. Wouldn't the percentages between the different materials affect the properties? Would such extremely distorted data make reasonable data sense? The authors must address this issue in Section 4.2.

3. The authors are requested to add comment #6 to the manuscript, it is well described.

4. The authors are requested to add comment #7 to the manuscript, which is a good demonstration of model efficacy. The prediction results can be credible only after the model has been proved to have good performance.
